# Effects of Al-Mg on the Microstructure and Phase Distribution of Zn-Al-Mg Coatings

**Ziyue Zhang** [1,2,*], **Jie Zhang** [2], **Xingyuan Zhao** [2], **Xuequn Cheng** [1], **Sheming Jiang** [2] **and Qifu Zhang** [2]

[1] Corrosion & Protection Center, University of Science & Technology Beijing, Beijing 100083, China
[2] National Engineering Laboratory of Advanced Coating Technology for Metals, Central Iron & Steel Research Institute, Beijing 100081, China
* Correspondence: zhangziyuelove@126.com

**Abstract:** In this work, the composition of the zinc–aluminum–magnesium alloy coating was designed to have a fixed aluminum–magnesium ratio of 1:1, while the content of aluminum and magnesium elements increases gradually within the range of 1–2 wt.%. The micro-morphology of the coating with different compositions was observed by a scanning electron microscope (SEM). Combined with the surface distribution results of energy dispersive spectrometer (EDS) analysis elements and the phase analysis results of diffraction of X-rays (XRD), the phase distribution of the coating is understood. The statistical calculation of the phase distribution was carried out after staining the SEM image by ImageJ, This is consistent with the solidification simulation results of the thermodynamic simulation software (PADAT). The influence of magnesium and aluminum elements on the microscopic morphology and phase distribution of the zinc–aluminum–magnesium (ZnAlMg) coating was studied, and the mechanism of action was analyzed. The results show that the volume ratio of binary eutectic phase ($Zn/MgZn_2$) and ternary eutectic phase ($Zn/Al/MgZn_2$) in the coating tends to increase as the contents of the two elements elevate. The quantity of $MgZn_2$ is the critical factor for the corrosion resistance of the coating; the more $MgZn_2$, the better the corrosion resistance.

**Keywords:** Zn-Al-Mg coatings; coating microstructure; phase distribution; corrosion resistance



## 1. Introduction

The corrosion of steel has caused huge commercial expenses and subsequently potential safety hazards with direct and indirect losses reaching about 4% GDP all over the world every year [1]. The hot-dip galvanizing surface treatment process, as the most cost-effective and feasible method for large-scale industrial production of steel corrosion prevention, is still not obsolete after nearly 200 years of development. With the continuous consumption of limited zinc resources and the deteriorating service environment of steel materials, pure Zn, Zn-Al and Al-Si cannot meet the demand in the status quo [2–5], so it is urgently needed to develop new high corrosion resistance and long-life hot-dip galvanized coatings.

Zinc–aluminum–magnesium (ZnAlMg), as the innovation of hot-dip plating technology in recent decades, has attracted more and more enterprises, universities and research institutes for its excellent corrosion resistance and edge cutting self-healing [6–10]. There is a relatively wide addition range of aluminum elements in ZnAlMg, which is mainly divided into three types: low aluminum (1~3 wt.%), medium aluminum (5~11 wt.%) and high aluminum (50~55 wt.%). The range of magnesium is small, with the 3 wt.% maximum addition, and a higher addition will sharply deteriorate the coating quality and cause splash spontaneous combustion of magnesium particles [11]. Compared with the traditional zinc coating, the corrosion resistance of ZnAlMg has been greatly promoted for the addition of magnesium and aluminum elements simultaneously. This is mainly attributed to the phase of the ZnAlMg, in which in addition to the zinc-rich phase, the $Zn/MgZn_2$ binary eutectic phase and $Zn/Al/MgZn_2$ ternary eutectic phase also ascend [12]. $MgZn_2$ improves the



corrosion resistance by leading to the formation of stable dense $Zn_5Cl_2(OH)_8 \cdot H_2O$, which prefers to hydrolyze in alkaline environments [13–15]. Meanwhile, the hardness of ZnAlMg increased for the addition of magnesium and it is assumed that magnesium can refine zinc grains and form $MgZn_2$ eutectic structures with a higher hardness [16,17].

Low-aluminum ZnAlMg products, which are mainly used in automobile and household electric appliances, require high performance stability and surface quality [18,19]. However, the element distribution in the molten zinc bath is not uniform, which is related to the properties of zinc, aluminum and magnesium, such as the oxidation tendency, density and alloy phase formation. So, the continuous dip process and feeding of alloy zinc ingots commonly fluctuate the practical volume ratio between zinc, aluminum and magnesium elements, as well as the composition and performance of same-batch products. In recent years, lots of scholars and experts have investigated the effects of aluminum or magnesium on the coating structure, alloy phase and corrosion resistance, but few focus on the effects of both aluminum and magnesium elements on the coating under the fixed volume ratio, which really reflects the actual production process. Under the condition that the Al/Mg ratio is 1:1 and the both element contents are in the range of 1–2 wt.%, the effect of Al and Mg on the structure, alloy phase and corrosion resistance of low aluminum ZnAlMg was studied in this work, which intended to provide theoretical basis and reference value for feeding of alloy ingots in the zinc molten bath.

## 2. Materials and Methods

The sample of ZnAlMg coating was prepared by using the self-developed hot-dip plating simulator (GCA-IV) in the laboratory(CISRI, NELACTM, China). This equipment simulates the conditions of the virtual hot-dip plating production line through flexibly adjusting the process parameters such as annealing temperature, immersion time and cooling rate. The experimental substrates were shaped in the dimensions of 120 (length) × 200 (width) × 0.7 mm (thickness) from commercial Interstitial-Free (IF) steel, then ultrasonic degreasing and cleaning in acetone as pretreatment processes were adopted. The main route is shown in Figure 1 and specific parameters are the following: the annealing temperature is 800 °C; zinc bath is kept at 450 °C; the steel plate at 460 °C immerses for 3 s. The coating thickness is controlled around 20 um by adjusting the $N_2$ gas knife flow rate. Finally, the sample after plating was cooled to room temperature by the rate of 5 °C/s Samples with 10 × 10 mm and 60 × 60 mm dimensions were cut from the experimental hot-dip plate and applied to the microscopic morphology observation and neutral salt spray test, respectively. The coating compositions are shown in Table 1.

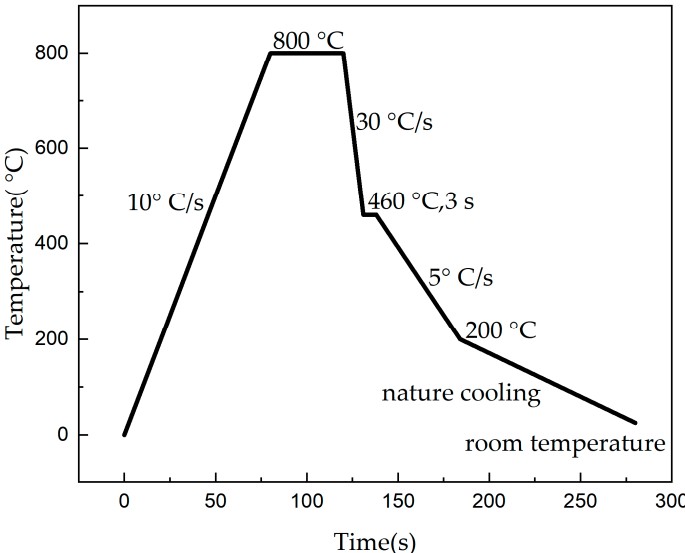

**Figure 1.** Experimental process control flow chart of hot-dip plating simulator.

**Table 1.** The compositions of experimental coatings.

| Coating (wt.%) | Al | Mg | Zn |
|---|---|---|---|
| Zn-1.0Al-1.0Mg | 1.05 | 1.04 | 97.91 |
| Zn-1.3Al-1.3Mg | 1.33 | 1.35 | 97.32 |
| Zn-1.6Al-1.6Mg | 1.61 | 1.64 | 96.75 |
| Zn-2.1Al-2.1Mg | 2.11 | 2.08 | 95.81 |

The FEI Quant 650-FEG field emission scanning electron microscope (SEM, FEI Inc., Valley City, ND, USA) equipped with energy dispersive spectroscope (EDS, EDAX Inc., Mahwah, NJ, USA) was utilized to characterize the microstructure and element distribution of the coating. A Brooke d8 advance X-ray diffractometer (Brooke Inc., Berlin, Germany) was used to carry out phase analysis of the coating through a Lynxeye XE detector (Brooke Inc., Berlin, Germany) and specific parameters were the following: Co target, tube current 40 mA, tube voltage 35 kV, scanning speed $2°$/min.

The solidification alloy phase of the coating was simulated by PANDAT software (CompuTherm Inc., Middleton, WI, USA). The samples were solidified at a cooling rate of $5\ °C$/s after plating, which is similar to the non-equilibrium solidification in fact. Therefore, the Scheil solidification model was adopted to simulate the solidification of the coating in this work.

The volume percentage of the phases was quantified by image analysis. ImageJ software (National Institutes of Health, Bethesda, MD, USA) was used to convert the surface/section SEM results into a binary format, and the threshold value was adjusted appropriately to obtain a clear boundary effect map of different phase structures, and then different alloys were colored according to different phases [20]. Then, the volume percentage of each phase structure was obtained by using the digital statistical algorithm of the software. The research studied the numerical statistics results of the two dimensions of the longitudinal depth and transverse surface of the coating at the same time, including the section statistics of the coating length of 5 mm, and the surface statistics of $5 \times 5$ mm area.

The dynamic polarization curve, corrosion potential (Ecorr) and corrosion current density (Icorr) were tested on the GAMRY Reference 600 electrochemical workstation (GAMRY Inc., Warminster, PA, USA). The special parameters were the following: the dynamic potential polarization solution is 5 wt.% sodium chloride solution (NaCl) solution, scanning range $-0.15\sim0.4$ V (relative to reference electrode of Pt), scanning speed 1 mV/s. The neutral salt spray test chamber was used to test the corrosion resistance and the experimental temperature of it was $35\ °C$. The NaCl concentration was $50 \pm 5$ g/L and the pH value of it reached 6.5–7.2 after atomization. The deposition amount was 1–2 mL/80 cm$^2$ h with the humidity greater than 95% Relative Humidity (RH). The test samples were taken out at early corrosion stage, when there is no red rust appearance. After removing the surface corrosion products, the surface morphology was observed through SEM for the comparison and analysis of microstructure and phase distribution of the coatings with different components.

## 3. Results and Analysis

### 3.1. Microstructure and Phase Composition

Figure 2 shows the SEM and EDS results of Zn-1.6Al-1.6Mg surface microstructure and the corresponding Zn, Al and Mg element distribution, respectively. In Figure 2a, it can be seen that there are mainly three distinct crystalline structures, including the Zn-hcp single phase uniformly distributed in spherical structure, and the binary and ternary eutectic phase in the shape of thick rod lamellar dendrites and thin rod lamellar dendrites, respectively. The XRD diffraction pattern of Zn-1.6Al-1.6Mg is shown in Figure 3b, which indicated that the diffraction peak of Zn phase is most intense and it is much higher than other peaks. The peaks of MgZn$_2$ and Al phases are generally low, but they still can be clearly seen. Bruycker et al. [12]. studied thermodynamic analysis on ZnAlMg and confirmed that ternary eutectic hcp Zn, fcc Al and MgZn$_2$ was formed at a certain temperature

of the composition of 93.6 wt.%Zn-3.9 wt.%Al-2.4 wt.%Mg. They reported that under rapid cooling conditions, $MgZn_2$ preferred to form, rather than the thermodynamically stable $Mg_2Zn_{11}$ phase. Similarly, only the $MgZn_2$ phase was found, in this study, to respond well [20–23].

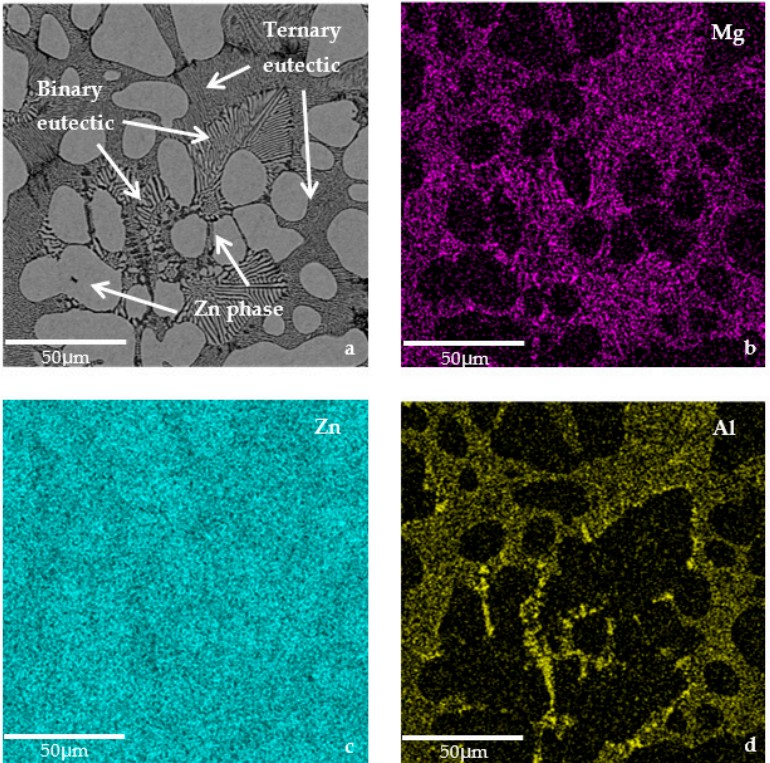

**Figure 2.** (**a**) SEM Surface Photos of Zn-1.6Al-1.6Mg coating and (**b**–**d**) EDS Analysis Results.

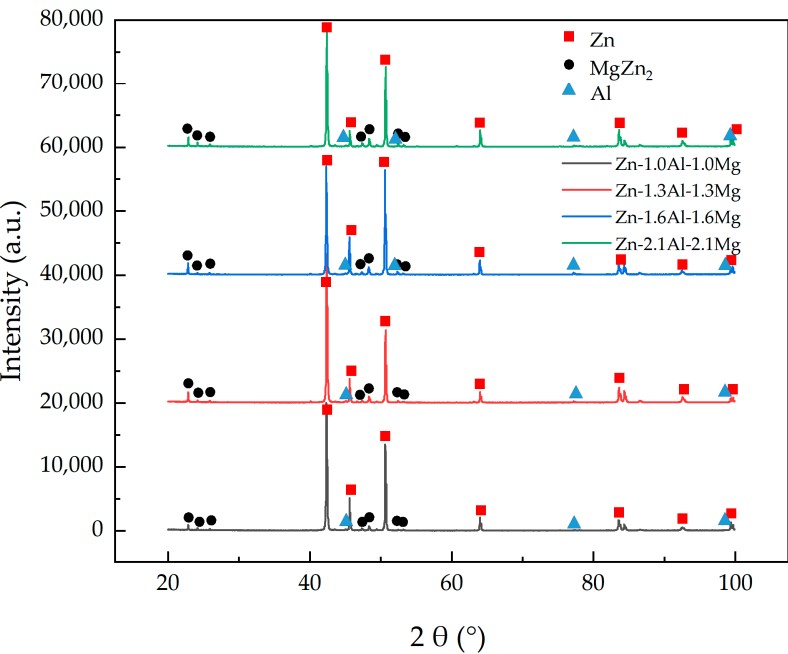

**Figure 3.** XRD Analysis of experimental coatings.

According to the element distribution in Figure 2b–d, Zn elements are distributed in the whole coating and only slightly reduced at the grain boundary. Meanwhile, Mg

elements are distributed in binary and ternary eutectic phases with Al elements only obviously arising in the ternary eutectic phase area. Corresponding to the XRD diffraction results in Figure 3, it confirmed that the spherical structure is the Zn phase, the coarse rod lamellar dendrite structure is the binary eutectic phase composed of Zn and $MgZn_2$, and the thin rod lamellar dendrite structure is the ternary eutectic phase composed of Zn, $MgZn_2$ and Al, which is consistent with the results in the literature [12,20].

The micro-morphology and phase composition of the coating directly affect its protective performance. In this study, the content of magnesium and aluminum is limited to 1–2 wt.%, and the content of Al and Mg is increased with the 1:1 Al/Mg ratio simultaneously. The scanning electron microscope photos of the coating cross section and surface section are shown in Figures 4 and 5, respectively. It can be seen that the microstructures with different components coatings are composed of a zinc-rich phase (Zn), binary eutectic phase (Zn, $MgZn_2$) and ternary eutectic phase (Zn, $MgZn_2$, Al). However, the volume ratio of each phase in the coating with different components is apparently different.

In order to quantitatively analyze the microstructures between different coatings, ImageJ software was used to dye the cross section and surface (Figures 4 and 5) and perform statistics on the volume ratio of various phases. The specific statistical results of cross section and surface are shown in Figure 6. The cross-section statistics show that the volume ratio of binary alloy phase increased gradually with the increase of the magnesium and aluminum contents, while the ternary alloy phase did not significantly fluctuate. In addition, the results of surface statistics displayed the similar trend of binary alloy phase with minor increase trend of ternary alloy phase.

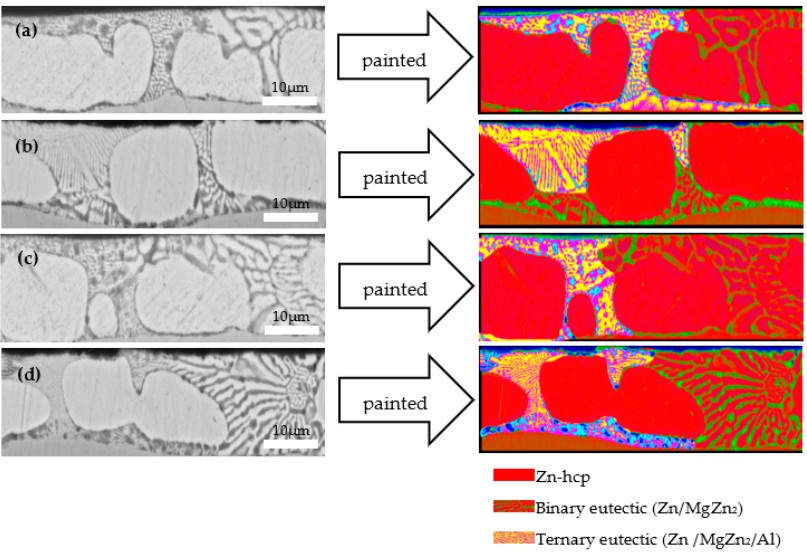

**Figure 4.** SEM photos of cross-section and ImageJ alloy phase dyeing photos of (**a**) Zn-1.0Al-1.0Mg, (**b**)Zn-1.3Al-1.3Mg, (**c**)Zn-1.6Al-1.6Mg, (**d**)Zn-2.1Al-2.1Mg.

The reason for the discrepancy of statistical results between the cross-section and surface may be attributed to the different cooling rates of the longitudinal depth during the nitrogen purging and cooling process. The whole solidification process belongs to the non-equilibrium state, which leads to the difference of the microstructure and phase distribution between non-equilibrium solidification state and equilibrium solidification state. The solidification process initiates from the surface, and the nucleation point nucleates and evolves from the surface to core, accompanied by gradual growth [1,24].

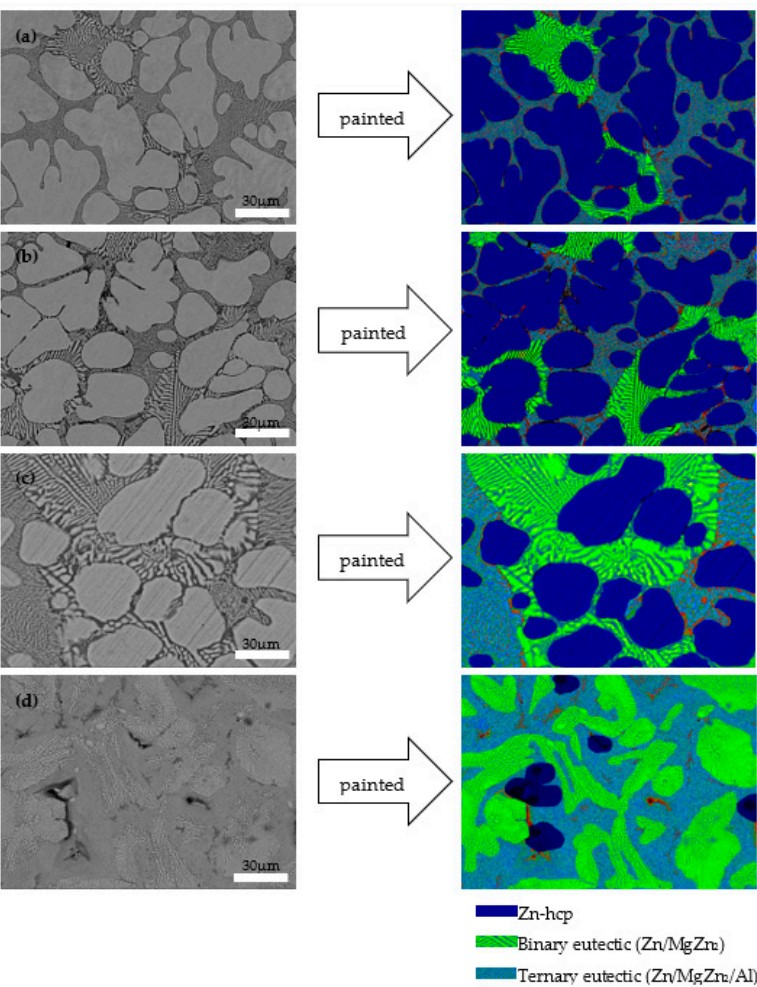

**Figure 5.** SEM photos of surface, and ImageJ alloy phase dyeing photos of (**a**) Zn-1.0Al-1.0Mg, (**b**) Zn-1.3Al-1.3Mg, (**c**) Zn-1.6Al-1.6Mg, (**d**) Zn-2.1Al-2.1Mg.

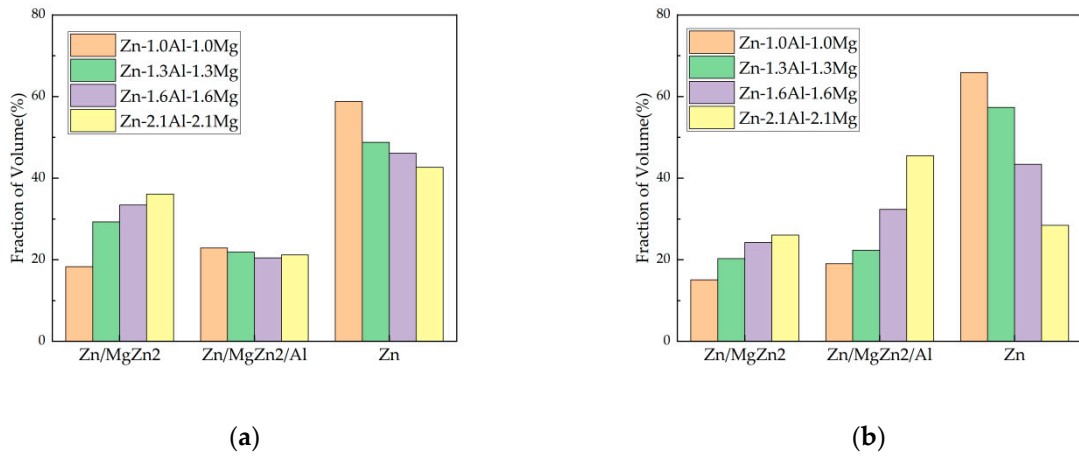

(**a**)  (**b**)

**Figure 6.** Statistical results of phase distribution of different coatings (**a**) cross-section, (**b**) surface section.

### 3.2. Thermodynamic Calculation and Solidification Simulation

Under the same cooling rate process conditions, the solidification curve of the ternary system varies with the variety of coating composition [24,25]. Solidification simulation calculation of four alloy components carried out with PANDAT software portrays the

schematic diagram of nonequilibrium solidification (Scheil) curve of ternary alloy coating, as shown in Figure 7. With the temperature decreasing, the liquid phase starts to precipitate a solid zinc-rich phase, and then a binary eutectic phase (Zn/MgZn$_2$) and ternary eutectic phase (Zn/MgZn$_2$/Al) appear successively. Along with the content of the two elements ascending from 1.0 to 2.1 wt.%, the precipitation temperature of the hcp Zn phase during solidification is gradually reduced, which also delays the initial precipitation of the binary eutectic phase, while the precipitation temperature of the ternary eutectic phase is consistent.

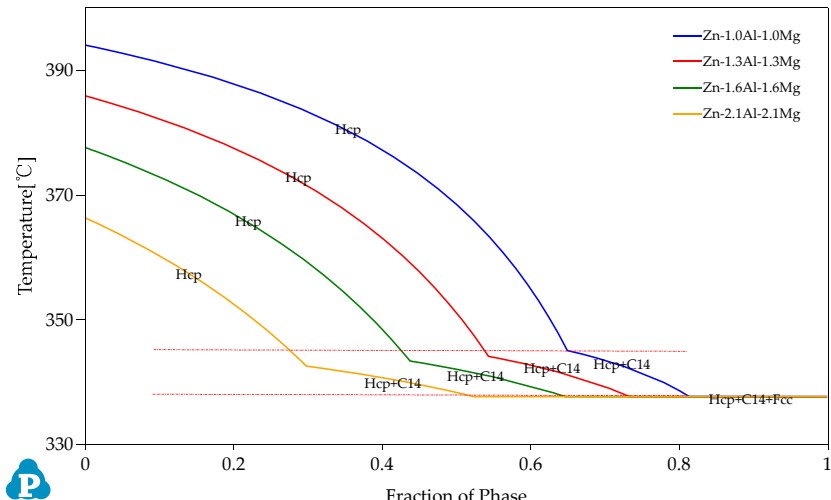

**Figure 7.** Non-equilibrium solidification (Scheil) curve of experimental coatings.

The solidification conditions and the final solidification structure distribution are obviously different even though the overall solidification process is alike. We can obtain the phase fraction of ternary system with different contents of aluminum and magnesium elements after non-equilibrium solidification by the database provided by the software as the calculation parameter. With the increase of element content, the binary and ternary alloy phase simultaneously increase in Figure 8a, which is consistent with the trend of the statistical results in Figure 6.

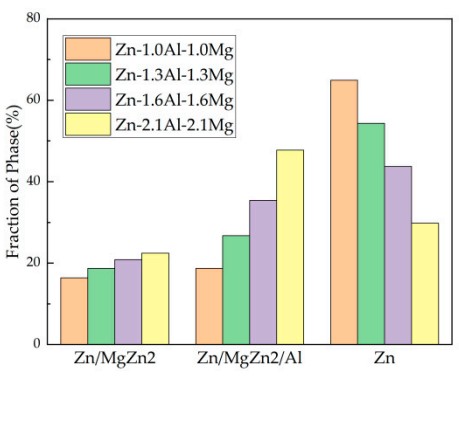

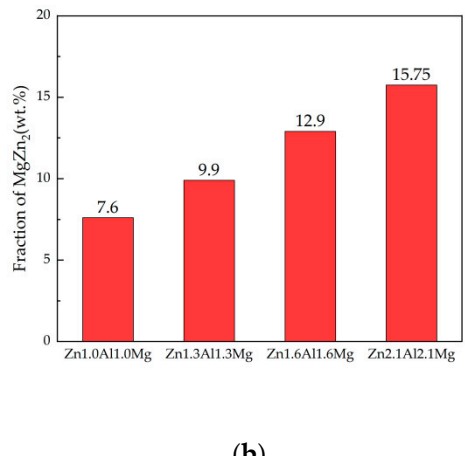

(**a**)

(**b**)

**Figure 8.** The results of PADAT-simulated non-equilibrium solidification; (**a**) phase fraction, (**b**) MgZn$_2$ fraction.

There are two reasons for this phenomenon: primarily, the alloy composition with higher element contents forms a binary eutectic layer relatively later (under the same cooling rate after plating), which means that MgZn$_2$ appears later. However, the temperature at

which the ternary eutectic phase appears is the same, which results in a shorter time for the formation of the binary eutectic phase. At the same time, according to the $MgZn_2$ fraction shown in Figure 8b, it also gradually increases with the increase of the element content. This shows that the binary eutectic reaction occurs faster in the coating with higher element content. As a result, the fluctuation of the phase fraction of the binary eutectic phase in different composition coatings is slight.

Secondly, since the aluminum only exists in the ternary eutectic phase, adding more aluminum will cause it to solidify in the ternary eutectic phase. Therefore, the addition of elements with the same aluminum–magnesium ratio will lead to a significant increase in the volume ratio of the ternary eutectic phase.

### 3.3. Corrosion Performance

In order to study the electrochemical behavior of the coating under the condition that the aluminum magnesium ratio is 1:1, the zinc aluminum magnesium coating was tested by dynamic polarization in 3.5% NaCl solution, and the corresponding curve is shown in Figure 9. The corrosion potential (Ecorr) and corrosion current density (icorr) were calculated by the Tafel method, as shown in Table 2. With the aluminum and magnesium content increasing from 1.0 wt.% to 2.1 wt.%, the corrosion potential (Ecorr) and current density (Icorr) decreased from $-1055$ mV to $-1120$ mV and from $5.68$ $\mu A/cm^2$ to $3.02$ $\mu A/cm^2$, respectively.

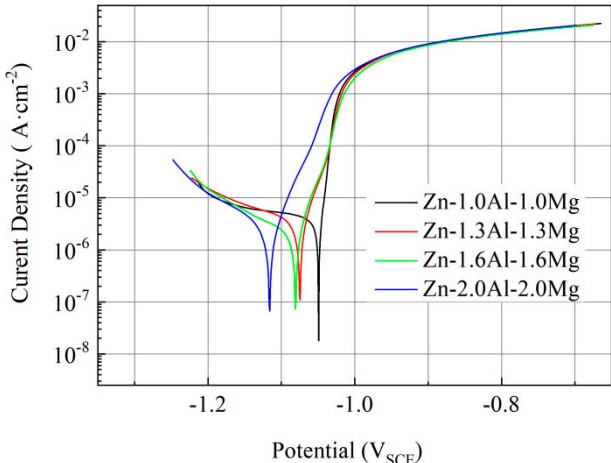

**Figure 9.** Dynamic polarization curve of experimental coatings in 3.5% NaCl solution.

**Table 2.** Corrosion potential and corrosion current density of experimental coating.

| Coatings (wt.%) | $E_{corr}$, mV$_{SCE}$ | $I_{corr}$, $\mu A/cm^2$ |
|---|---|---|
| Zn-1.0Al-1.0Mg | $-1055 \pm 5$ | $5.68 \pm 0.5$ |
| Zn-1.3Al-1.3Mg | $-1070 \pm 5$ | $4.71 \pm 5$ |
| Zn-1.6Al-1.6Mg | $-1185 \pm 5$ | $3.95 \pm 5$ |
| Zn-2.1Al-2.1Mg | $-1120 \pm 5$ | $3.02 \pm 5$ |

The results were mainly attributed to the fact that the potentials of both Al and $MgZn_2$ are more negative than that of Zn (Figure 9). In addition, there is galvanic corrosion between Zn and $MgZn_2$, and the potential difference between them is 0.5 ($V_{SCE}$) according to the literature [26]. $MgZn_2$ is dissolved and provided $Mg^{2+}$ to react with $OH^-$ to form the precipitation of $Mg(OH)_2$. The $Mg(OH)_2$ leads to the decrease of pH value, thus creating thermodynamic conditions for the formation of water-insoluble simonkolleite ($Zn_5Cl_2(OH)_8 \cdot H_2O$) to prevent its further hydrolysis in alkaline environment [27]. The added aluminum element mainly participates in the formation of $Mg_6Al_2(OH)_{16}CO_3 \cdot 4H_2O$ and $Zn_2Al(OH)_6(CO_3)_{1/2} \cdot xH_2O$) in the corrosion process, which,

as protective corrosion products, inhibit the further corrosion rate, thus improving the corrosion resistance [28,29]. With the increase in the aluminum and magnesium content, the volume ratio of binary eutectic and ternary eutectic phases increases, which leads to the corrosion potential and corrosion current density of the coating being reduced. During the corrosion process, the increase of binary eutectic and ternary eutectic phases leads to a greater formation of $Zn_5Cl_2(OH)_8\cdot H_2O$ in the corrosion products. In the range of 1–2 wt.% Mg and Al, the coating with higher Al and Mg content has better sacrificial protection for steel plates [4,30].

During the neutral salt spray corrosion test, the SEM of the sample surface showed that $MgZn_2$ in the interdendritic area was preferentially dissolved, as shown in Figure 10. With the increase of aluminum and magnesium content in the coating, the volume ratio of binary eutectic and ternary eutectic phases in the coating increased. At the same corrosion time, the area of $MgZn_2$ dendrite corrosion in the coating with high aluminum and magnesium content decreased, indicating that more binary eutectic and ternary eutectic phases reduce the corrosion rate and improve the corrosion resistance of the coating.

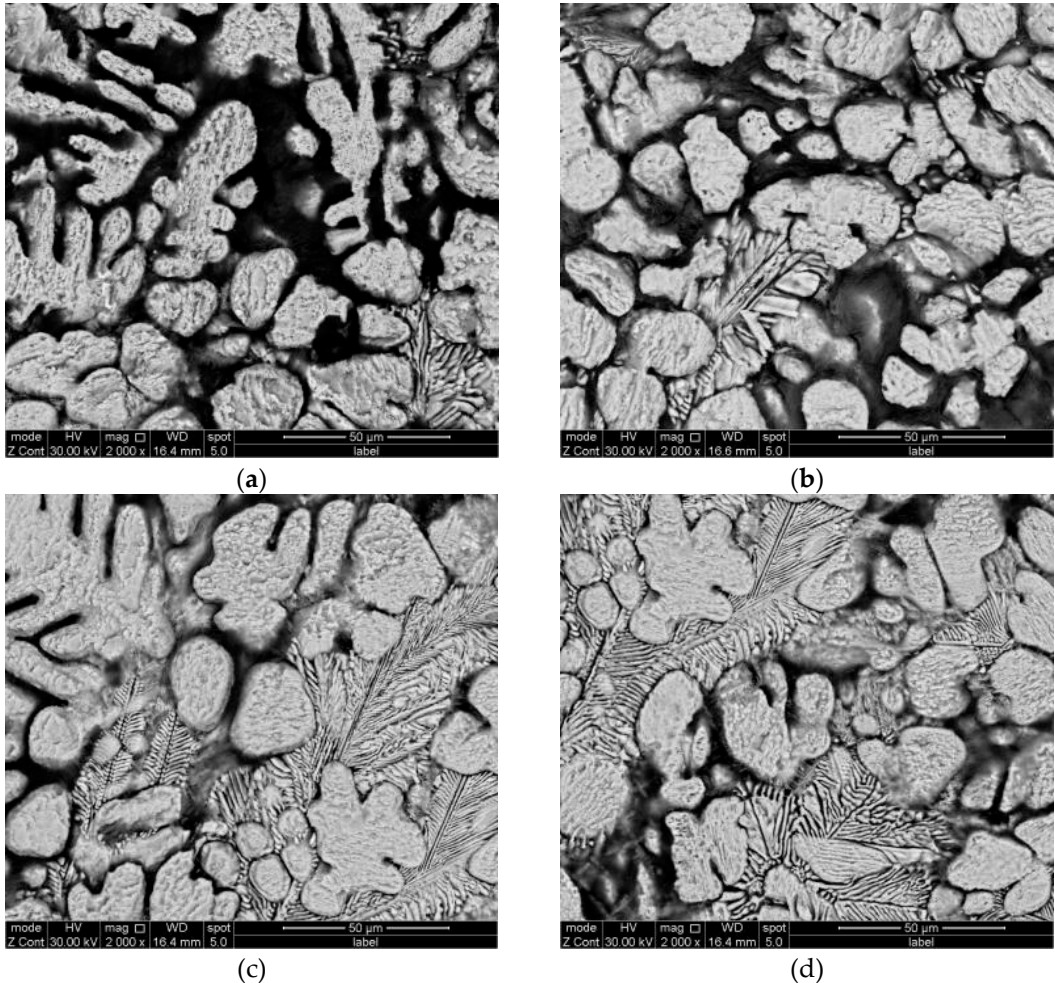

**Figure 10.** The SEM results of the experiment coatings, removing corrosion products after 15 weeks of a cyclic salt spray test' (**a**) Zn-1.0Al-1.0Mg, (**b**) Zn-1.3Al-1.3Mg, (**c**) Zn-1.6Al-1.6Mg, (**d**) Zn-2.1Al-2.1Mg.

## 4. Conclusions

(1) Under the condition that the ratio of aluminum to magnesium of zinc aluminum magnesium coating is 1:1, and the content of aluminum and magnesium is in the range of 1–2 wt.%, with the increase of the two elements, the micro morphology of the coating has a certain regularity of change, in which the composition of the

phase type has not changed and is composed of a zinc-rich phase, binary ($Zn/MgZn_2$) and ternary eutectic phase ($Zn/MgZn_2/Al$). The zinc-rich phase decreases with the synchronous increase of magnesium and aluminum, and the volume ratio of binary and ternary eutectic phases increases.

(2) The calculation results of the thermodynamic simulation software are consistent with the statistical results of the cross section, and surface SEM images of the zinc aluminum magnesium coating show the trend of the coating phase distribution, basically. The content of two elements increases at the same time, resulting in an obvious increase in the volume ratio of ternary eutectic phase $Zn/MgZn_2/Al$. In addition, with the increase of magnesium and aluminum, $MgZn_2$ shows an obvious increasing trend, and the presence of aluminum in the ternary eutectic structure is the main reason for this phase distribution trend.

(3) The quantity of $MgZn_2$ in the coating is the key factor for its corrosion resistance. The research shows that with the increase of element contents in the coating, the corrosion resistance of the coating will also be improved. The main reason is that the existence of $MgZn_2$ reduces the corrosion potential of the coating, preferentially dissolves in the corrosion process, and plays a role in the protection of sacrificial anode.

**Author Contributions:** Data curation, Z.Z. and J.Z.; Methodology, X.C., Q.Z. and S.J.; Software, Z.Z. and J.Z.; Writing—review & editing, Z.Z. and X.Z. All authors have read and agreed to the published version of the manuscript.

**Funding:** This research received no external funding.

**Data Availability Statement:** Not applicable.

**Acknowledgments:** This research was carried out with the help of the 'Environmentally friendly high corrosion and haze resistant coatings' Project, supported by NELACTM of China. This support is gratefully acknowledged.

**Conflicts of Interest:** The authors declare no conflict of interest.

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
