# Peer review of "Effects of Al-Mg on the Microstructure and Phase Distribution of Zn-Al-Mg Coatings"

_metals, doi:10.3390/met13010046_

Round 1

Reviewer 1 Report

1.       First, the manuscript is written in very poor English. In particular, some sentences are either unclear or have grammar errors. Besides, the authors unnecessarily use many times capital instead of small letters. Also, they frequently do not put a space between a value and a unit, between the point at the end of a sentence, etc., etc. The paper cannot be accepted till its English is fundamentally improved. I attach the manuscript with errors shown by color.

2.       The reviewed manuscript is limited to only one variable: a joint content of Al+Mg at a mass ratio of both elements 1:1. Please justify such a design as it gives no possibility to distinguish the effect of Al and Mg, separately.

3.       The investigations are limited to microstructure (SEM, AFM, XRD) and corrosion studies. The potentiodynamic curves do not show the straight lines long at least for one decade on a logarithmic scale. Thus, the determined by software corrosion parameters are very approximate and the tests should be repeated at least 3-5 values till the standard deviation will be less than 25% of the mean value. However, I see no SDs, has been only one corrosion measurement for any alloy?

4.       Give the company delivering elements and their densities.

5.       What does it mean, IF steel?

6.       The proper XRD apparatus commercial name is D8ADVANCE and not d8advance, check it.

7.       What reference electrode has been used?

8.       Panited or Painted (see figures 4 and 5)?

9.       Line 202 and others “proportion of the phase”; not a proportion, but a volume ratio.

Author Response

Dear professor:
Thanks for your professional and meticulous comments,here are my explanations to your questions:

1 I have carefully revised every sentence and word you marked as red in the paper,I'm so sorry to take up too much time on such low-level writing errors.

2 According to the literature, it is known that the content of aluminum and magnesium both have an impact on the performance of the coating, but in industrial production, the contents of two elements fluctuate simultaneously. It is difficult to reach the element proportion which can ensure the quality stability of the coated steel plate in the alloy solution only by considering the supplement of a single element. The purpose of this study is to understand the effect of the same Mg/Al ratio on the coating performance within a certain element content range. Next, there will be further research on the alloy coating with different Mg/Al ratios.Finally, it is hoped that the corresponding relationship between the coating performance and the ratio of magnesium to aluminum can be obtained.

3 I have modified the question you raised in the attachment Word. Three groups of samples were made for each group of polarization curve. I took the average value only in the table before the modification, and modified SDs in the text.

4 This company has a project cooperation with us. They don't want the brand of the product  to appear in the paper, so I can only give the category name of the material.

5 Interstitial-Free(IF) steel .

6 Modification completed in paper.

7 reference electrode of Pt ,Modification completed in paper.

8 Modification completed in paper.

9 Modification completed in paper.I think the  "volume ratio"s statement is more accurate than "proportion"thanks for your Valuable comments.

Reviewer 2 Report

The subject is of high technological and scientific interest. Data are worth from a technological point of view, the manuscript is well and clearly written, the figures and tables are of enough quality and presents new work.

Page 4, Line 109. "The deposition amount is 1-2mL/80cm2 · h with..." Please change cm2 from sub index to super index 

Page 8. Figure 6. caption" The SEM results of experimental coating surface section and corresponding ImageJ alloy 169 phase dyeing photos (a)Zn-1.0Al-1.0Mg,(b)Zn-1.3Al-1.3Mg,(c)Zn-1.6Al-1.6Mg,(d)Zn-2.1Al-2.1Mg." It does not correspond with the Figure 6 only (a) and (b)

Author Response

Dear professor:

Thanks for your professional comments,and the Line 109 and the Figure 6 has been modification completed.

Reviewer 3 Report

The article demonstrates the effect of Al+Mg equiatomic additions in hot-dip Zn-Al-Mg coating on steel on the microstructure and its corrosion resistance. Zinc and Zn-Al-Mg coated steel sheets are the most important for the automotive, building and household appliance industry. The anti-corrosion barrier properties of such coatings depend on differences in the electrochemical potentials of the individual phases in 3-component Zn-Al-Mg system. The authors of the article demonstrated that Al+Mg additives radically change the microstructure and phase composition of coatings. The main conclusion is based on the fact that the proportion of MgZn2 in the coating is the key factor for its corrosion resistance. The architecture of the article is extremely well built. The set of modern experimental techniques is minimal and sufficient. Conclusions are justified.

Among the technical comments, it should be noted that the Caption for Figure 6 does not correspond to the presented bar charts. It needs to be corrected.

Author Response

Dear professor:

Thanks for your professional comments,and the Figure 6 has been modification completed.
